# A Dense Mapping Algorithm Based on Spatiotemporal Consistency

**DOI:** 10.3390/s23041876

**Published:** 2023-02-07

**Authors:** Ning Liu, Chuangding Li, Gao Wang, Zibin Wu, Deping Li

**Affiliations:** 1Robotics Intelligence Technology Research Institute, Jinan University, 601 Huangpu Avenue West, Guangzhou 510632, China; 2College of Information Science and Technology, Jinan University, 601 Huangpu Avenue West, Guangzhou 510632, China; 3School of Intelligent Systems Science and Engineering, Jinan University, 206 Qianshan Road, Zhuhai 519070, China

**Keywords:** dense mapping, local map extraction, spatiotemporal consistency, point cloud fusion

## Abstract

Dense mapping is an important part of mobile robot navigation and environmental understanding. Aiming to address the problem that Dense Surfel Mapping relies on the input of a common-view relationship, we propose a local map extraction strategy based on spatiotemporal consistency. The local map is extracted through the inter-frame pose observability and temporal continuity. To reduce the blurring of map fusion caused by the different viewing angles, a normal constraint is added to the map fusion and weight initialization. To achieve continuous and stable time efficiency, we dynamically adjust the parameters of superpixel extraction. The experimental results on the ICL-NUIM and KITTI datasets show that the partial reconstruction accuracy is improved by approximately 27–43%. In addition, the system achieves a greater than 15 Hz real-time performance using only CPU computation, which is improved by approximately 13%.

## 1. Introduction

Simultaneous Localization and Mapping (SLAM) [1] is a critical technology. It is important for mobile robots to be able to locate and construct maps in unfamiliar environments autonomously. A mobile robot’s map reconstruction ability plays a crucial role in recognizing its 3D environment, navigating safely, and completing tasks [2].

Existing mature SLAM frameworks mainly include keyframe-based and mapping-based reconstruction methods. The former is more flexible in management, and the latter can achieve higher precision. Keyframe-based frameworks focus on localization. These frameworks have become mainstream because the positioning algorithm they employ can achieve real-time requirements. However, the map obtained by directly overlaying point clouds is usually not sufficiently accurate. Mapping-based frameworks, on the other hand, take accurate maps as the main goal and basically require a GPU for acceleration. The research direction of real-time 3D reconstruction is developing towards the reconstruction of large-scale scenes. However, there are still bottlenecks in terms of reconstruction accuracy, real-time performance, and adaptability to the environment. These bottlenecks are due to the physical characteristics of RGB-D sensors and the limitations of computing resources. In 2017, Wang et al. proposed that a usable reconstructed map for mobile robot applications should satisfy the following: (1) The map can densely cover the environment to provide sufficient environmental information for the robot; (2) The system has good scalability; (3) The system has good global consistency; (4) The system can fuse different sensors and depth maps of different quality. To meet the above requirements, Dense Surfel Mapping [3] was proposed. The algorithm is based on the surfel model, which extracts superpixels [4] from the depth and intensity images to model the surfel and applies depth images of different qualities. The resulting map achieves global consistency thanks to the fast map deformation [3]. Most importantly, the algorithm can work in real time with only CPU computation.

However, Dense Surfel Mapping has the following problems: (1) The lack of general-purpose ability of local map extraction: the extraction relies on the covisibility graph of ORB-SLAM2 [5], and pose estimation algorithms without covisibility graphs can only extract based information based on time series. Thus, we extract the local map based on the pose relationship between frames. This eliminates the dependence on the covisibility graph of the input, and it makes the input simpler and the system more versatile. (2) Simple weighted average fusion may lead to inaccuracy in the surfels with a better viewing angle. We add normal constraints to the surfel weight’s initialization. Surfels with better view angles will be initialized with greater weights. For surfels with large differences from normal, we only keep the one with the better viewing angle instead of using weighted average fusion. This improves the reconstruction accuracy. (3) The superpixel extraction traverses the entire image. It is unnecessary to handle the regions with invalid depth or beyond the maximum mapping distance, so we filter out the invalid regions before performing superpixel extraction, and we dynamically adjust the parameters of the superpixel extraction based on spatial continuity and temporal stability. Thanks to the dynamic superpixel extraction, the time efficiency of the system are further improved.

In summary, the main contributions of this paper are the following.

We propose a local map extraction and fusion strategy based on spatiotemporal consistency. The local map is extracted through the inter-frame pose observability and temporal continuity. This eliminates the dependence on the common-view relationship of the pose estimation algorithm and is suitable for various pose estimation algorithms.A dynamic superpixel extraction. We dynamically adjust the parameters of superpixel extraction based on spatial continuity and temporal stability, achieving continuous and stable time efficiency.The normal constraints are added to the surfel weight initialization and fusion so that surfels with better viewing angles are kept during map fusion.The experimental results on the ICL-NUIM dataset show that the partial reconstruction accuracy is improved by approximately 27–43%. The experimental results on the KITTI dataset show that the method proposed in this paper is effective. The system achieves a greater than 15Hz real-time performance, which is an improvement of approximately 13%.

## 2. Related Work

This section mainly introduces the development of dense reconstruction methods and their scalability and efficiency.

With the commercialization of RGB-D sensors such as Kinect [6], a 3D reconstruction based on RGB-D sensors gradually attracted the attention of researchers, steadily developing and maturing. At present, dense mapping methods are mainly divided into voxel-based methods [7,8,9,10], surfel-based methods [3,11], and so on. KinectFusion [12] realized real-time 3D reconstruction based on an RGB-D camera for the first time. This system uses the TSDF (truncated signed distance function) [13] model to reconstruct the environment, but it takes a lot of memory to store the voxel grid. ElasticFusion [14] is a rare reconstruction model using the surfel model [15] model, which focuses on the fine construction of the map. ElasticFusion also improves the pose estimation and reconstruction accuracy by continuously optimizing the reconstructed map. However, ElasticFusion is only suitable for small scenes because of the large computation required. BundleFusion [16] achieves detailed local surface detail registration using the sparse-to-dense registration strategy and achieves real-time continuous model updates using the re-integration model update strategy. It is currently one of the best algorithms for dense 3D reconstruction based on an RGB-D camera. In recent years, many researchers have focused on the combination of neural networks and 3D reconstruction techniques. NICE-SLAM [17] used a hierarchical neural implicit encoding to reconstruct large-scale scenes. Guo et al. used neural implicit representation to model the scene with the Manhattan-world constraint [18]. Azinović et al. effectively incorporated the TSDF model in the NeRF framework [19]. SimpleRecon [20] tried to learn the depth map directly by using an encoder–decoder architecture based on cost volume, and it introduced metadata into the cost volume to provide more prior knowledge for model training. BNV-Fusion [21] proposed a bi-level fusion algorithm to achieve superior performance. The above reconstruction algorithms need GPU acceleration to achieve good real-time performance because of the huge amount of calculation required. Wang et al. proposed a novel mapping system named Dense Surfel Mapping [3]. The system can fuse sequential depth maps into a globally consistent model in real time without GPU acceleration. Because of the novel superpixel model, the system is suitable for room-scale and urban-scale environments.

The scalability of the voxel-based method is general. It requires a lot of memory to store voxels, so the voxel-based method is, therefore, not suitable for large-scale scenarios, such as KinectFusion [12]. Kintinuous [7] uses a cyclical buffer to improve the scalability of the mapping system. Nießner et al. [22] proposed a voxel hashing method that only stores reconstructed sparse surfaces. This method greatly improves the model’s scalability. Compared with voxel-based methods, surfel-based methods are more scalable. This is because surfel-based systems only store reconstructed surface point clouds. Dense Surfel Mapping and [23] further improve scalability by maintaining local maps. Dense Surfel Mapping [3] extracts local maps according to the common-view relationship provided by the ORB-SLAM2. Similar to Dense Surfel Mapping, we use a more general local map extraction method to improve the scalability. The method eliminates the model’s dependence on the input. It is extracted through the inter-frame pose observability and temporal continuity. It is more versatile and can be compatible with various pose estimation algorithms.

Runtime efficiency is an essential indicator of the mapping algorithm. Different algorithms offer unique methods to improve runtime efficiency. Voxblox [9], based on voxels, proposes grouped raycasting: each point is projected to a voxel, all points in the same voxel are averaged, and only one raycasting process is performed to speed up fusion. FIESTA uses Indexing Data Structures and Doubly Linked Lists for map maintenance [10]. The efficient data structures and BFS framework of FIESTA allow the system to update as few nodes as possible. Steinbrucker et al. [24] represent scenes using an octree, which is an efficient way to store 3D surfaces. FlashFusion [25] filters out invalid chunks using valid chunk selection; that is, only the chunks in the frustum of the camera view are considered. This highly efficient method allows the algorithm to render at 25 Hz. Dense Surfel Mapping [3] uses superpixels to extract surfels quickly. A local map is maintained to reuse the existing surfels and reduce the memory burden. We further filter out the invalid regions of the image and dynamically adjust the parameters of the superpixel extraction. Thanks to the dynamic superpixel extraction method, our system achieves better time efficiency.

## 3. System Overview

As shown in Figure 1, the system framework is mainly divided into five parts.

### 3.1. System Input

The system input is mainly divided into two parts: one is the depth and RGB image obtained by the RGB-D sensor, and the other is the pose graph obtained by the pose estimation algorithms (e.g., ORB-SLAM series [5,26,27], VINS-Mono [28], VINS-Fusion [29]). The pose graph in [3] is similar to the covisibility graph of ORBB-SLAM2. It includes the path and the common-view relationships of the keyframes because it needs the covisibility graph to extract the local map. The input of a pose graph is complex, so it cannot be widely used in various pose graph inputs. Different from [3], the pose graph used in this paper is just the path of keyframes or the ordinary frames. It is simpler and more generic for pose estimation algorithms, and the constraints are relatively loose.

### 3.2. Global Consistency Deformation

Same as [3], if the input pose graph is updated, the previous poses are optimized. The map is quickly deformed according to the pose difference between the current pose graph and the database. Surfels attached to frame *F* are deformed according to the matrix $T_{2}T_{1}^{-1}$, where $T_{1}$$\;\in R^{4\times 4}$ is the pose of the frame *F* in the database and $T_{2}$$\;\in R^{4\times 4}$ is the pose of the frame *F* in the current pose graph. Then, $T_{1}$ is replaced by $T_{2}$ and stored in the database. The pose is a homogeneous transformation matrix that includes a rotation matrix and a translation vector.

### 3.3. Superpixel and Local Map Extraction

In [3], superpixels are extracted by a *k*-means approach adapted from the extended SLIC [30]. Pixels are clustered [31] according to their intensity, depth, and pixel location. Finally, a down-sampled superpixels image is obtained. The superpixel extraction in [3] traverses the entire image. It is unnecessary to handle the regions with invalid depth or beyond the maximum mapping distance. So, as shown in Figure 1, we first filter out the invalid regions before the superpixel’s extraction. Meanwhile, we dynamically adjust the parameters of the superpixel extraction based on spatial continuity and temporal stability. This allows the system to achieve better time efficiency. More details are described in Section 4.2. The local map extraction in [3] is based on the covisibility graph of the input. Keyframes with the number of minimum edges to the current keyframe below $G_{\delta }$ are locally consistent. Surfels attached to these keyframes are extracted as the local map [3]. To make the system more versatile, we simplify the input in Section 3.1, and we propose a spatiotemporal consistent local map extraction strategy that is simple and effective. We extract the local map based on the pose relationship between frames and continuity in time. More details are described in Section 4.1.

### 3.4. Map Fusion

In this part, extracted surfels in the local map are fused with extracted surfels in the current frame. The work of [3] transforms the local surfels into the current frame. A weighted average is used to fuse the transformed surfel, and the surfel is extracted in the current frame with a similar depth and normals. However, simple weighted average fusion may lead to inaccurate surfels with better viewing angles. We, thus, add the normal constraints to the surfel weight initialization so that a surfel with a better view angle will be initialized with a greater weight. For surfels with a large difference in normals, we directly keep the one with the better viewing angle instead of performing weighted average fusion. This improves the accuracy of the surfels. And more details are described in Section 4.3.

### 3.5. Map Publication

In this part, the publication is an independent thread. We retrieve the reconstructed map from the database regularly and publish it as a ROS topic. The topic can be subscribed to for use in later applications, such as navigation and planning.

## 4. Methods and Principles

### 4.1. Spatiotemporally Consistent Local Map Extraction

Reconstructing large-scale environments may generate millions of surfels. To reduce map growth, local maps are extracted to reuse and fuse previous surfels and redundant surfels. In this paper, we extract the relevant common-view frames as a local map based on the pose relationship between the two frames. As shown in Figure 2, the pose relationship between two frames is mainly divided into three cases.

#### 4.1.1. In the Same Direction Horizontally

As shown in Figure 2a,b, two frames ($F_{1}$ and $F_{2}$) are nearly parallel. The distance between the two frames is calculated as:(1)$$D=\|p_{1}-p_{2}\|\;,$$
where $p_{1}$$\;\in R^{3}$ and $p_{2}$$\;\in R^{3}$ are the 3D coordinates of frames $F_{1}$ and $F_{2}$. The cosine of the angle between the two frames’ directions is determined as:(2)$$\cos\alpha =\frac{n_{1}\cdot n_{2}}{\|n_{1}\left\|\right\|n_{2}\|}\;,$$
where $n_{1}$$\;\in R^{3}$ and $n_{2}$$\;\in R^{3}$ are the direction vectors of frames $F_{1}$ and $F_{2}$, respectively. The constraints should satisfy: (1) the distance *D* between two frames is less than the maximum mapping distance k·far_dist, where *k* is the scale factor, and (2) their angle α is less than the camera’s field of view (FOV), denoted as $\theta_{th}$. There is a common area between two frames only when constraints (1) and (2) are both satisfied.

#### 4.1.2. In the Same Direction or Opposite

As shown in Figure 2d,e, frames $F_{1}$ and $F_{2}$ are in forward or opposite motion. The coordinates of $F_{1}$ are projected to the coordinate system of $F_{2}$, and the pixel coordinates are calculated as follows:(3)$$\begin{array}{c} {\left[{p_{1\_F_{2}}}^{T},\;1\right]}^{T}=T_{wF_{2}}^{-1}{\left[{p_{1\_w}}^{T},\;1\right]}^{T}\;, \end{array}$$(4)$$\begin{array}{c} {}[u_{1},v_{1}{,1]}_{F_{2}}^{T}=Kp_{1\_F_{2}}\;, \end{array}$$
where $T_{wF_{2}}$$\;\in R^{4\times 4}$ is the pose matrix of the frame $F_{2}$ in global coordinates, *K*$\;\in R^{3\times 3}$ is the camera intrinsic matrix, and $p_{1\_w}$$\;\in R^{3}$ is the 3D global coordinate of the frame $F_{1}$. Similarly, the coordinates of $F_{2}$ are projected to the coordinate system of $F_{1}$ and the pixel coordinates are calculated as follows:(5)$$\begin{array}{c} {\left[{p_{2\_F_{2}}}^{T},\;1\right]}^{T}=T_{wF_{1}}^{-1}{\left[{p_{2\_w}}^{T},\;1\right]}^{T}\;, \end{array}$$(6)$$\begin{array}{c} {}[u_{2},v_{2}{,1]}_{F_{1}}^{T}=Kp_{2\_F_{1}}\;, \end{array}$$
where $T_{wF_{1}}$$\;\in R^{4\times 4}$ is the pose matrix of the frame $F_{1}$ in global coordinates, *K*$\;\in R^{3\times 3}$ is the camera intrinsic matrix, and $p_{2\_w}$$\;\in R^{3}$ is the 3D global coordinate of the frame $F_{2}$. $F_{1}$ ’s pixel coordinates ${\left[u_{1},v_{1}\right]}_{F_{2}}^{T}$$\;\in R^{2}$ are in the valid coordinate range of the image ($V^{2\times 1}$$\;\in R^{2}$). This means that $u_{1}$ is between 0 and the image’s width, while $v_{1}$ is between 0 and the image’s height. The depth $p_{1\_F_{2}}{|}_{z}$ is less than the maximum mapping distance k·far_dist. $F_{1}$ and $F_{2}$ are considered to have a common-view area when the above two conditions are satisfied, and it is the same for $F_{2}$. Surfels attached to this frame can be used as local map frames.

#### 4.1.3. Back to Back

As shown in Figure 2c, the directions of frames $F_{1}$ and $F_{2}$ are almost opposite. There is no overlap in the fields of view between them. The projection of each frame is not within the other’s field of view, and the direction angle is greater than $\theta_{th}$. In general, this case does not satisfy Section 4.1.1 and Section 4.1.2 at the same time. In this case, the two frames have no common area and cannot be used as local map frames.

#### 4.1.4. Summary

In summary, the current frame $F_{j}$ and extracted frames $F_{i}$ should satisfy:(7)$$\left\{\begin{array}{c} \|p_{i}-p_{j}\|\;<k\cdot far\_dist\\ \cos^{-1}\frac{n_{i}\cdot n_{j}}{\|n_{i}\left\|\right\|n_{j}\|}<\theta_{th}\\ or\\ \left(K\;\left(T_{wF_{i}}^{-1}{\left[{p_{j\_w}}^{T},1\right]}^{T}\right){|}_{3\times 1}\right){|}_{2\times 1}\in V^{2\times 1}\\ \left(T_{wF_{i}}^{-1}{\left[{p_{j\_w}}^{T},1\right]}^{T}\right){|}_{z}\;<k\cdot far\_dist\\ or\\ \left(K\;\left(T_{wF_{j}}^{-1}{\left[{p_{i\_w}}^{T},1\right]}^{T}\right){|}_{3\times 1}\right){|}_{2\times 1}\in V^{2\times 1}\\ \left(T_{wF_{j}}^{-1}{\left[{p_{i\_w}}^{T},1\right]}^{T}\right){|}_{z}\;<k\cdot far\_dist\;, \end{array}\right.$$
where $V^{2\times 1}$$\;\in R^{2}$ is the valid coordinate range of the image. To further enhance the temporal continuity of the local map, frames that are continuous in time are also extracted. For a value of $F_{i}$ that satisfies the above constraints, 2n frames in the time-series $\left\{F_{i-n},F_{i-n+1},\cdots ,F_{i-1},F_{i+1},\cdots ,F_{i+n-1},F_{i+n}\right\}$ are continuously extracted as the local map at the same time.

The complete algorithm is shown in Algorithm 1.
**Algorithm****1.** Local Map Extraction.**Input****:** *j* is the index of the current frame. $T_{wF_{j}}$ is the pose of the current frame. poseDatabase is the pose database that stores the poses of each frame and their surfels. far_dist is the maximum mapping distance.**Output****:** localIndexes is a vector of the local frame indexes. localSurfels is a vector of the local surfels.1:localIndexes.CLEAR()2:localSurfels.CLEAR()3:**for each** $F_{i}\in \;poseDatabase$**do**4:    flag←false5:    $T_{F_{j}F_{i}}$← transform($T_{wF_{i}}$, $T_{wF_{j}}$)6:    $\left[u,v\right]$← project($T_{F_{j}F_{i}}$)7:    **if** isValidRange($\left[u,v\right]$) && $T_{F_{j}F_{i}}{|}_{z}$≤k·far_dist **then**8:        flag←true9:    **end if**10:    $T_{F_{i}F_{j}}$← transform($T_{wF_{j}}$, $T_{wF_{i}}$)11:    $\left[u,v\right]$← project($T_{F_{i}F_{j}}$)12:    **if** isValidRange($\left[u,v\right]$) && $T_{F_{i}F_{j}}{|}_{z}$≤k·far_dist **then**13:        flag←true14:    **end if**15:    **if** distance($T_{wF_{i}}$, $T_{wF_{j}}$) ≤k·far_dist && angle($T_{wF_{i}}$, $T_{wF_{j}}$) ≤$\theta_{th}$ **then**16:        flag←true17:    **end if**18:    **if** flag **then**19:        **for** t←−n,−n+1...n−1,n **do**20:           localIndexes.PUSH(i+t)21:        **end for**22:    **end if**23:**end for**24:**for each** i∈localIndexes**do**25:    localSurfel.INSERT(poseDatabase[*i*].surfels)26:**end for**

### 4.2. Dynamic Superpixel Extraction

Reconstructing large-scale scenes puts a large burden on memory. Superpixels can solve this problem well. Similar to [3], the superpixels are extracted from the intensity and depth images.

The cluster center is described as $C_{i}={\left[x_{i},y_{i},d_{i},c_{i},r_{i}\right]}^{T}$, where ${\left[x_{i},y_{i}\right]}^{T}$ is the average location of clustered pixels and $d_{i}$$\;\in R^{+}$, $c_{i}$$\;\in R^{+}$, and $r_{i}$$\;\in R^{+}$ are the average depth, intensity value, and the radius of the superpixel, respectively. Each pixel *u* is assigned to a cluster center according to the distance *D* between itself and its neighborhood cluster center $C_{i}$ as follows:(8)$$\begin{array}{c} D=\frac{{\left(x_{i}-u_{x}\right)}^{2}+{\left(y_{i}-u_{y}\right)}^{2}}{N_{s}^{2}}+\frac{{\left(c_{i}-u_{i}\right)}^{2}}{N_{c}^{2}}+\frac{{\left(1/d_{i}-1/u_{d}\right)}^{2}}{N_{d}^{2}}\;, \end{array}$$
where ${\left[u_{x},u_{y},u_{d},u_{i}\right]}^{T}$ are the location, depth, and intensity of pixel *u*. $N_{s}$, $N_{c}$, and $N_{d}$ are used for normalization. This is the same as in [3].

To enhance the time efficiency of the superpixel extraction, we only handle the depth-valid pixels in the assignment. The superpixel size sp_size and the maximum mapping distance far_dist are the main parameters that affect the time efficiency. We periodically resize the superpixels in time-series frames with the high common-view area:(9)$$\begin{array}{c} sp\_size_{i+1}=\left\{\begin{array}{cc} SP\_SIZE, & e_{acc}\geq FAR\_DIST\;or\;e_{rot}\geq c_{1}\\ c_{2}\cdot SP\_SIZE, & sp\_size_{i}\geq c_{2}\cdot SP\_SIZE\\ sp\_size_{i}+1, & others \end{array}\right.\;, \end{array}$$
where SP_SIZE and FAR_DIST are the basic superpixel size and maximum mapping distance, $c_{1}$ is the rotation difference threshold (default is 0.1), $c_{2}$ is the scale constant, and e_rot$\;\in R^{+}$ and e_acc$\;\in R^{+}$ are the rotation errors and the accumulated pose errors, respectively, between two consecutive frames. The maximum mapping distance far_dist is dynamically adjusted according to the real-time efficiency as follows:(10)$$\begin{array}{c} far\_dist_{i+1}=\left\{\begin{array}{cc} c_{3}\cdot far\_dist_{i}, & k\leq -c_{4}\\ far\_dist_{i}/c_{3}, & k\geq c_{4}\\ far\_dist_{i}, & others \end{array}\right.\;, \end{array}$$
where $c_{3}$ (default is 1.1) is the scale factor, $c_{4}$ (default is 3) is a positive integer. *k* means that the time cost of consecutive ∣k∣ frames is lower than the average time cost when *k* is negative and the time cost of consecutive ∣k∣ frames is higher than the average time cost when *k* is positive.

### 4.3. Projection Matching and Optimal Observation Normal Map Fusion

There will be a large number of redundant surfels between the surfels generated by the current frame and the local map because of the similar poses. The same surfels observed in different orientations of the frame should be fused to reduce map growth. In this paper, the same surfels are matched by projection and then culled and fused according to their position and normal constraints.

Different from the surfel in [3], the surfel in this paper is $S={\left[S_{p},S_{n},S_{c},S_{w},S_{i},S_{t},S_{v},S_{r}\right]}^{T}$, where $S_{p}$$\;\in R^{3}$ is the global coordinate, $S_{n}$$\;\in R^{3}$ is the unit normal, $S_{c}$$\;\in R^{+}$ is the color vector, $S_{w}$$\;\in R^{+}$ is the weight coefficient, $S_{i}$∈N is the frame number to which it belongs, $S_{t}$∈N is the number of updates, $S_{v}$$\;\in R^{+}$ is the observation cosine in frame $S_{i}$, and $S_{r}$ is the radius of the surfel. An observation cosine is added for the screening of better observation surfels. Project the surfel $S^{j}$ in the local map to the coordinate system of the current frame $F_{i}$: (11)$$\begin{array}{c} {\left[{S_{p\_i}^{j}}^{T},\;1\right]}^{T}=T_{wi}^{-1}{\left[{S_{p\_w}^{j}}^{T},1\right]}^{T}\;, \end{array}$$(12)$$\begin{array}{c} S_{n\_i}^{j}=R_{wi}^{-1}S_{n\_w}^{j}\;, \end{array}$$
where $S_{p\_i}^{j}$$\;\in R^{3}$ and $S_{n\_i}^{j}$$\;\in R^{3}$ are the 3D coordinates and normals of $S^{j}$ in the coordinate system of the current frame, and $T_{wi}$$\;\in R^{4\times 4}$ is the pose matrix of the current frame $F_{i}$. $R_{wi}$$\in R^{3\times 3}$ is the rotation matrix of $F_{i}$.

As shown in Figure 3a, the red squares are surfels generated by the current frame, and the dots are surfels of the local map. Surfels can be divided into three categories based on the relationship between surfels of the local map and the newly generated ones:

1. Outlier surfels, such as the blue dots in Figure 3a, whose projections are not within the field of view of the current frame:(13)$${\left[u_{j},v_{j}\right]}^{T}=\left(KS_{p\_i}^{j}\right){|}_{2\times 1}\;\notin V^{2\times 1}\;,$$
where *K*$\;\in R^{3\times 3}$ is the camera intrinsic matrix, $V^{2\times 1}$$\;\in R^{2}$ is the valid coordinate range of the image, or the projection depth is much larger than the depth of the corresponding surfel $S^{i}$ in the current frame:(14)$$\begin{array}{c} S_{p\_i}^{j}{|}_{z}-S_{p\_i}^{i}{|}_{z}\;>th\;, \end{array}$$
(15)$$\begin{array}{c} th=\min\left\{min\_th,\;\frac{{\left(S_{p\_i}^{j}{|}_{z}\right)}^{2}\cdot \sigma }{b\cdot f\cdot k\cdot S_{v}^{j}}\right\}\;, \end{array}$$
where th is the depth difference threshold of outliers, set to 0.5m in the first culling and calculated by the Formula (Equation 15) in secondary culling, min_th is the minimum threshold constant, *b* is the baseline of the camera, *f* is the focal length of the camera, σ is the parallax standard deviation, *k* is the scale factor of the observed cosine (default is 1.5). The equation shows that there will be a larger tolerance threshold with the farther distance and the larger viewing angle. Thus, the farther surfels are considered more lenient for fusion. Surfels that satisfy condition (Equation 13) or do not satisfy condition (Equation 14) are not considered for fusion.

2. Conflict surfels, such as the gray dots in Figure 3a, satisfy ${\left[u_{j},v_{j}\right]}^{T}\in V^{2\times 1}$. If its depth difference is less than −th, then these surfels are considered to be conflicting and need to be replaced.

3. Update surfels, such as the black dots in Figure 3a, satisfy ${\left[u_{j},v_{j}\right]}^{T}\notin V^{2\times 1}$ after projection, and its depth difference is within ±th. These surfels are considered to be similar to the corresponding newly generated surfels and must be fused and updated to reduce map growth. After the projection constraint, a normal constraint is applied to the matching local map surfel $S^{j}$ and the newly generated $S^{i}$:(16)$$S_{n}^{j}\cdot S_{n}^{i}>v_{th}\;,$$
where $v_{th}$ defaults to 0.9. If the matching surfels are not satisfied with the constraint from (Equation 16), a strategy based on the best view angle is applied to reserve better surfels. As shown in Figure 4, pose1 and pose2 observe the same superpixel sp. Compared with pose2, which is easily affected by reflection and inaccurate depth, pose1 observes it in a better view. Because there is a smaller angle with the normal, pose1 obtains a high-quality depth and normal that better describes the superpixel.

In summary, the results of surfel fusion are shown in Figure 3b. The weighted average fusion with normal constraints of the matching surfels is as follows: (17)$$\begin{array}{c} S_{p}^{j}\leftarrow \left\{\begin{array}{cc} \frac{S_{w}^{j}\cdot S_{p}^{j}+S_{w}^{i}\cdot S_{p}^{i}}{S_{w}^{j}+S_{w}^{i}}\;, & S_{n}^{j}\cdot S_{n}^{i}\geq v_{th}\\ S_{p}^{j}\;, & S_{n}^{j}\cdot S_{n}^{i}<v_{th},\;S_{v}^{j}>S_{v}^{i}\\ S_{p}^{i}\;, & S_{n}^{j}\cdot S_{n}^{i}<v_{th},\;S_{v}^{j}\leq S_{v}^{i} \end{array}\right.\\ S_{n}^{j}\leftarrow \left\{\begin{array}{cc} \frac{S_{w}^{j}\cdot S_{n}^{j}+S_{w}^{i}\cdot S_{n}^{i}}{S_{w}^{j}+S_{w}^{i}}\;, & S_{n}^{j}\cdot S_{n}^{i}\geq v_{th}\\ S_{n}^{j}\;, & S_{n}^{j}\cdot S_{n}^{i}<v_{th},\;S_{v}^{j}>S_{v}^{i}\\ S_{n}^{i}\;, & S_{n}^{j}\cdot S_{n}^{i}<v_{th},\;S_{v}^{j}\leq S_{v}^{i} \end{array}\right.\\ S_{r}^{j}\leftarrow \left\{\begin{array}{cc} \min(S_{r}^{i},S_{r}^{j})\;, & S_{n}^{j}\cdot S_{n}^{i}\geq v_{th}\\ S_{r}^{j}\;, & S_{n}^{j}\cdot S_{n}^{i}<v_{th},\;S_{v}^{j}>S_{v}^{i}\\ S_{r}^{i}\;, & S_{n}^{j}\cdot S_{n}^{i}<v_{th},\;S_{v}^{j}\leq S_{v}^{i} \end{array}\right.\\ S_{w}^{j}\leftarrow \left\{\begin{array}{cc} S_{w}^{j}+S_{w}^{i}\;, & S_{n}^{j}\cdot S_{n}^{i}\geq v_{th}\\ S_{w}^{j}\;, & S_{n}^{j}\cdot S_{n}^{i}<v_{th},\;S_{v}^{j}>S_{v}^{i}\\ S_{w}^{i}\;, & S_{n}^{j}\cdot S_{n}^{i}<v_{th},\;S_{v}^{j}\leq S_{v}^{i} \end{array}\right.\\ S_{c}^{j}\leftarrow \left\{\begin{array}{cc} S_{c}^{j}\;, & S_{v}^{j}>S_{v}^{i}\\ S_{c}^{i}\;, & others \end{array}\right.\\ S_{v}^{j}\leftarrow \max\left(S_{v}^{i},S_{v}^{j}\right)\\ S_{t}^{j}\leftarrow S_{t}^{j}+1\\ S_{i}^{j}\leftarrow S_{i}^{i} \end{array}\;.$$

Considering the inaccuracy caused by distant observations and oblique observations, the weight coefficient of initialization $S_{w}$ is related to the depth and observation cosine:(18)$$S_{w}=\min\left(1,\frac{k\cdot S_{v}}{d}\right)\;,$$
where *d* is the depth of *S* in the current camera coordinate system. Because our input of the pose graph is loose, only the paths of the keyframes or ordinary frames are needed. For ordinary frame reconstruction, especially in large-scale scenes, the rate of pose estimation is high. Surfels whose last update was 15 frames ago and which have been updated less than five times are considered outliers and will be removed. Of course, this is not suitable for reconstruction with a low pose estimation rate input.

## 5. Experiments

This section mainly evaluates the algorithm through public datasets. The algorithm’s accuracy is evaluated using the ICL-NUIM dataset [32] and compared with other state-of-the-art algorithms such as Dense Surfel Mapping [3], ElasticFusion [14], BundleFusion [16], and FlashFusion [25]. The local consistency and the time efficiency in large-scale environments are evaluated using the KITTI odometry dataset [33].

The platform used to evaluate our method is a four-core, 4G memory Ubuntu18.04 system configured by VMware under an AMD Ryzen5 4600H. To maintain the same conditions as the comparison method, we also use ORB-SLAM2 in RGB-D mode to track the camera motion and provide the pose graph.

### 5.1. ICL-NUIM Reconstruction Accuracy

The ICL-NUIM [32] dataset is a synthetic virtual dataset provided by Imperial College London and the National University of Ireland. It is designed to evaluate RGB-D, visual odometry, and SLAM algorithms and is compatible with the TUM dataset. The dataset mainly includes two scenes: a living room and an office room. In addition to the ground truth, the living room scene also has a 3D surface ground truth [32]. It is perfectly suited not just for benchmarking camera trajectories but also reconstruction. To simulate real-world data, the dataset adds noise to both RGB images and depth images.

This experiment uses the living room scene with noise to evaluate the reconstruction accuracy of the algorithm. The input image resolution is 640 × 480. A superpixel size of SP_SIZE=4, FAR_DIST=3m is used for surfel fusion. The mean error of the reconstruction results is calculated using the CloudCompare tool:(19)$$MAE=\frac{1}{n}\cdot \sum_{i=1}^{n}\left\|p_{i}-{\hat{p}}_{i}\right\|\;,$$
where $p_{i}$ is the 3D coordinate of the reconstructed point cloud, and $\hat{p_{i}}$ is the closest true value of the 3D surface to $p_{i}$. The experimental results are compared with algorithms such as Dense Surfel Mapping [3], ElasticFusion [14], BundleFusion [16], and FlashFusion [25].

The reconstruction map and the corresponding error heat map are shown in Figure 5. The accuracy evaluation results are shown in Table 1.

Among the algorithms in Table 1, both ElasticFusion and BundleFusion require GPU acceleration. FlashFusion and Dense Surfel Mapping can be directly run in real-time under the CPU. Based on Dense Surfel Mapping, our method can also be run in real-time without GPU acceleration. In terms of reconstruction accuracy, our accuracy of kt0 reaches 0.4 cm, which is slightly higher than the 0.5 cm of BundleFusion, and the accuracy of kt3 also reaches 0.8 cm, which is the same as BundleFusion. Compared with Dense Surfel Mapping, the accuracy of our method is slightly higher in kt0 and kt2, the same in kt3, and slightly worse in kt1.

As shown in Figure 5, the reconstruction point clouds of sofas and murals are clear, and even the text in them is faintly visible. It can be seen from the heat map that the main errors are concentrated within 1 cm. There are some errors around 1 cm of kt1, mainly on the walls on both sides of the z-axis. These are mainly caused by the inaccurate pose estimation of ORB-SLAM2. There is always a unidirectional deviation of 2 cm–3 cm between the estimated pose and the ground truth in the y-axis and z-axis. This also reflects the side that the algorithm has a certain tolerance for error in pose estimation. It can be seen in Figure 5a that the error of the walls is small. This is because the walls of kt0 have been reconstructed from the front. This is a wonderful perspective for observing the object. According to the strategy presented in Section 4.3, these surfels will have a great weight in the fusion, and even surfels reconstructed from the front will directly replace surfels in the local map instead of weighted average fusion. This is also the case in kt2. This also explains why the accuracies of kt0 and kt2 are improved in Table 1.

### 5.2. Kitti Reconstruction Efficiency

This section mainly shows the method’s reconstruction performance in large-scale environments. The KITTI dataset is a computer vision algorithm evaluation dataset created by the Karlsruhe Institute of Technology (KIT) and the Toyota University of Technology at Chicago (TTIC) for autonomous driving scenarios. The dataset mainly contains large outdoor scenes such as urban areas, villages, and highways. The KITTI odometry used in this section mainly consists of 22 binocular sequences, 11 of which (00–10) have real trajectories. Here we only use the sequence 00.

The classic PSMNet [34] depth prediction neural network is used to predict depth images using binocular images. This is because the KITTI odometry does not provide depth images. To verify the spatiotemporally consistency of the local map extraction and fusion method proposed in this paper, we use the ground truth trajectories provided by the dataset directly.

The reconstruction results are shown in Figure 6. The left shows the motion trajectory of the camera, and the right is the map reconstructed by our method in real-time. The reconstructed map covers all the areas that the camera passes through without problems, such as large-scale blurring and disappearance.

Figure 7 shows the local detail of the reconstruction selected from the red box area in Figure 6b, which is a revisited area. The left is the result with a local map extraction only based on time series, and the right is the result of our method. The reconstructed cars in the red box on the left appeared misaligned, and the right solves the problem. As can be seen from Figure 6, it takes hundreds of frames to pass through the red box area twice. The left in Figure 7 fails to extract the previous surfels for fusion. The error of the pose when reconstructing two frames leads to ghosting. The right side of the figure extracts the first reconstructed surfels as a local map for fusion so that there is no such problem. It can, thus, be seen that our method of local map extraction and fusion performs well on the consistency of the local map.

The memory usage of the surfels throughout the runtime is shown in Figure 8. The orange curve is the result of our method without removing outliers. The black one is the result of our method with removing outliers. The blue one is the result of extracting local maps only based on time-series. There is almost no difference in the first 3200 frames because the car was moving to an unknown area in the scene. Between about 3200 and 4000 frames, the memory usage of our method stays almost unchanged because the car revisits the area between two red flags in Figure 6a, but the blue curve is still growing In addition, it can be seen that the memory usages of the black curve and orange curve are quite different. That is because large-scale scenes can easily generate outliers, and the input pose graph rate is high (10 Hz). If the outliers are not removed, the number of reconstructed surfels will greatly increase. Of course, when the rate of the input pose graph is low, the strategy of removing outliers is not advisable, causing the normal surfels to be removed and resulting in an incomplete reconstruction scene.

As shown in Figure 9 and Table 2, as the superpixel’s size becomes smaller, the average time cost per frame increases. As the maximum mapping distance increases, the average time cost per frame increases, too. This is because we filter the invalid pixels and only handle the valid regions. When SP_SIZE = 8 and FAR_DIST = 20, the average time cost is around 60 ms per frame, making our system about 15 Hz in real-time. Compared with [3], our time efficiency is improved by approximately 13% under the same conditions (SP_SIZE = 8 and FAR_DIST = 30).

To verify the effect of $c_{3}$ and $c_{4}$ of Formula (Equation 10) on time efficiency, we control the values of $c_{3}$ and $c_{4}$ in our experiments. SP_SIZE = 8 and FAR_DIST = 30 are used in the experiments. The results are shown in Figure 10 and Figure 11 and Table 3. With the increase in $c_{3}$ and $c_{4}$, large jitters in the running time occasionally appear, which have a larger standard deviation. This is because a larger $c_{4}$ will cause a delay in the dynamic adjustment parameters that will not be adjusted in time according to the current running state. A larger $c_{3}$ results in a larger change in far_dist, which is not conducive to smooth and stable time efficiency.

## 6. Conclusions

Aiming to improve the generalization ability of Dense Surfel Mapping, we propose a spatiotemporally consistent local map extraction method. It makes the system widely applicable to various pose estimation algorithms that only need to provide the path of poses. Meanwhile, the system achieves local accuracy and local consistency. An optimal observation normal fusion strategy is used for better surfels fusion. Compared with [3], the partial reconstruction accuracy of ICL-NUIM is improved by approximately 27–43%. Thanks to the dynamically adjusted superpixel extraction strategy, we achieve a greater than 15 Hz real-time performance. This is 13% higher than [3]. The mapping system is suitable for room-scale and large-scale environments. The local map reuses the previous surfels in space so that the memory usage grows according to the environment’s scale instead of the runtime. Adjusting superpixels according to their time cost makes the runtime more stable and efficient. The system achieves a balance between memory usage and time efficiency.

## Figures and Tables

**Figure 1 sensors-23-01876-f001:**
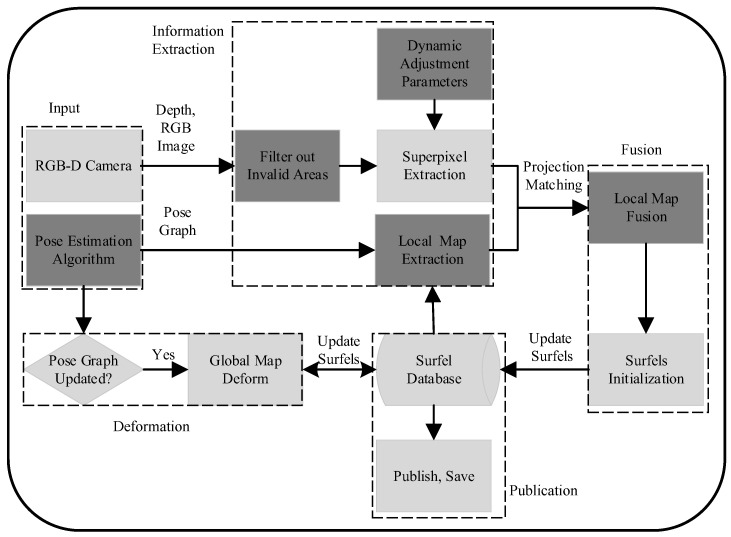
System framework. The system is mainly divided into five parts, as shown by the dotted boxes.

**Figure 2 sensors-23-01876-f002:**
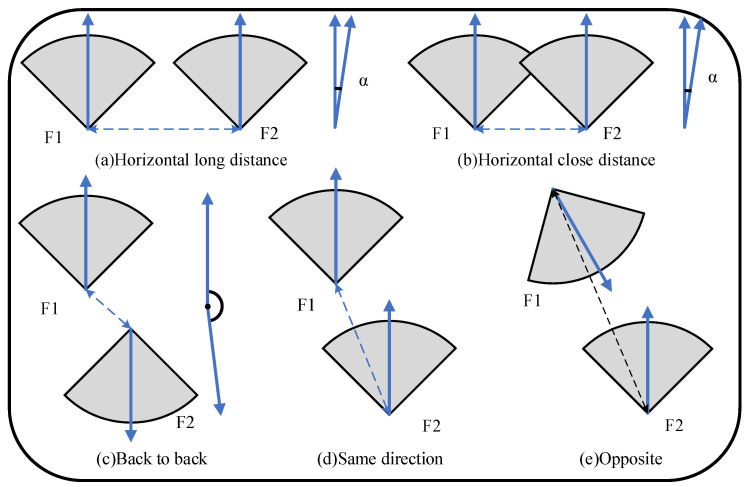
Inter-frame pose relationship. The gray sectors represent the view of the camera, and the arrows in the sectors are their directions.

**Figure 3 sensors-23-01876-f003:**
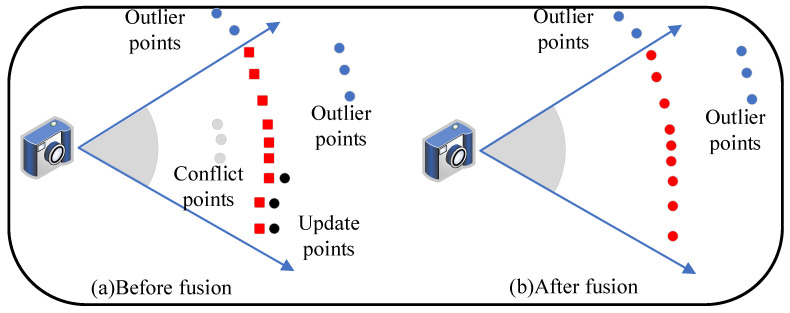
Data association. New generated surfels (red squares) are fused with surfels (dots) of the local map.

**Figure 4 sensors-23-01876-f004:**
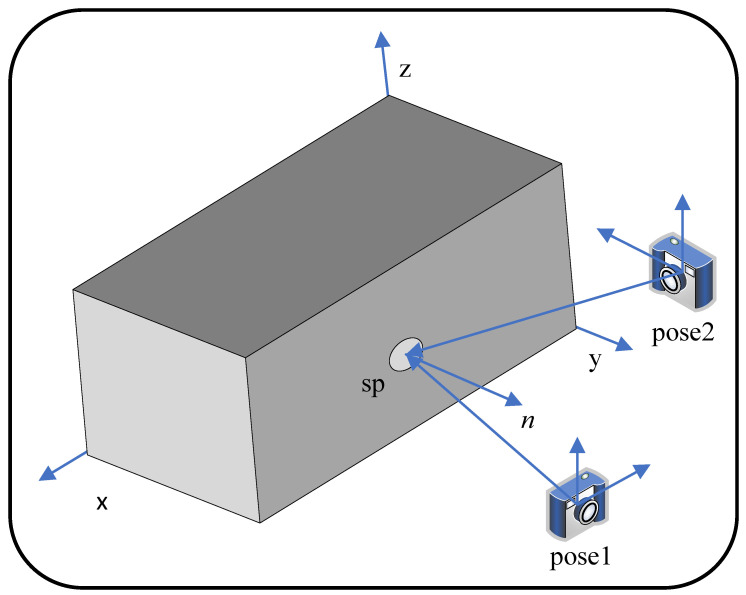
Best viewing angle. sp is a superpixel, and *n* is its normal. Pose1 observes sp in a better view because of its smaller angle with the normal.

**Figure 5 sensors-23-01876-f005:**
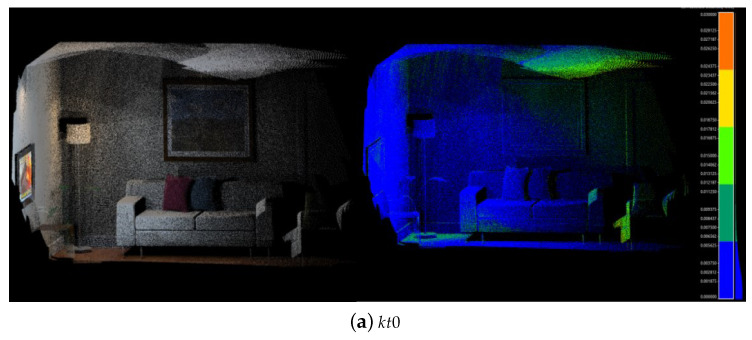
Reconstruction results of the ICL-NUIM dataset. The left shows the reconstructed point clouds. The right shows the error heatmap, in which the red part represents a 3 cm error and the blue part represents a 0 cm error.

**Figure 6 sensors-23-01876-f006:**
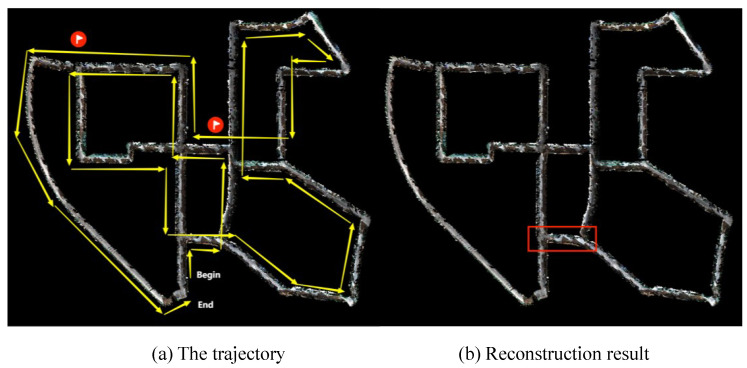
Reconstruction results of KITTI odometry sequence 00. (**a**) Motion trajectory of the camera. (**b**) Point clouds of reconstruction. The area in the red frame of (**b**) is the revisit area.

**Figure 7 sensors-23-01876-f007:**
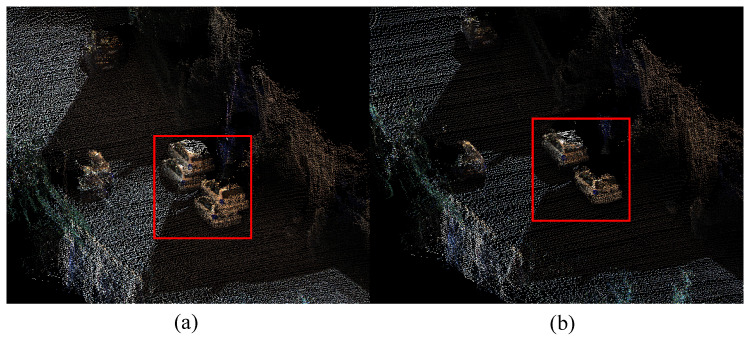
Reconstruction details of the revisited area. (**a**) Reconstruction details with a local map extraction based on time-series. (**b**) Reconstruction details of our method. The red frames show the performance differences between the two methods. Compared with (**b**), the cars of (**a**) are misaligned.

**Figure 8 sensors-23-01876-f008:**
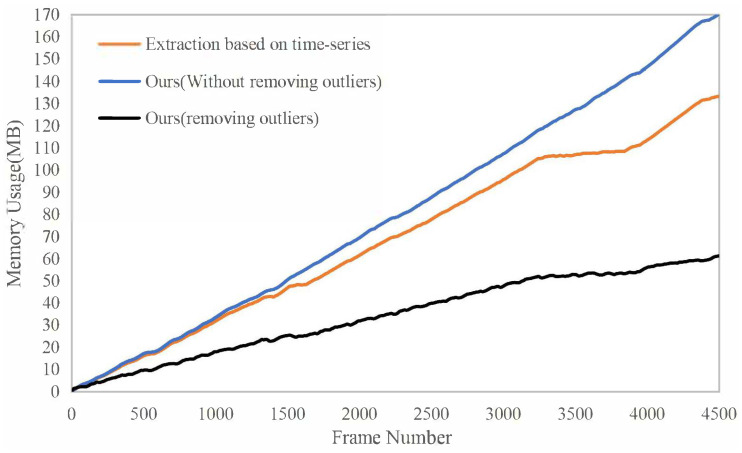
Memory usage of reconstructing KITTI odometry sequence 00. Between about 3200 and 4000 frames, the memory usage of our method stays almost unchanged.

**Figure 9 sensors-23-01876-f009:**
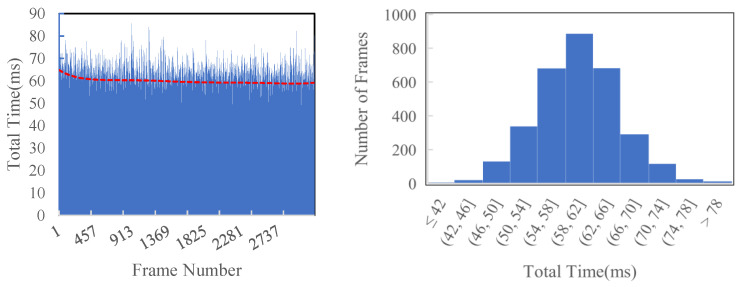
Time efficiency of reconstructing KITTI odometry sequence 00 (SP_SIZE = 8, FAR_DIST = 20 m). The average cost time is around 60 ms per frame.

**Figure 10 sensors-23-01876-f010:**
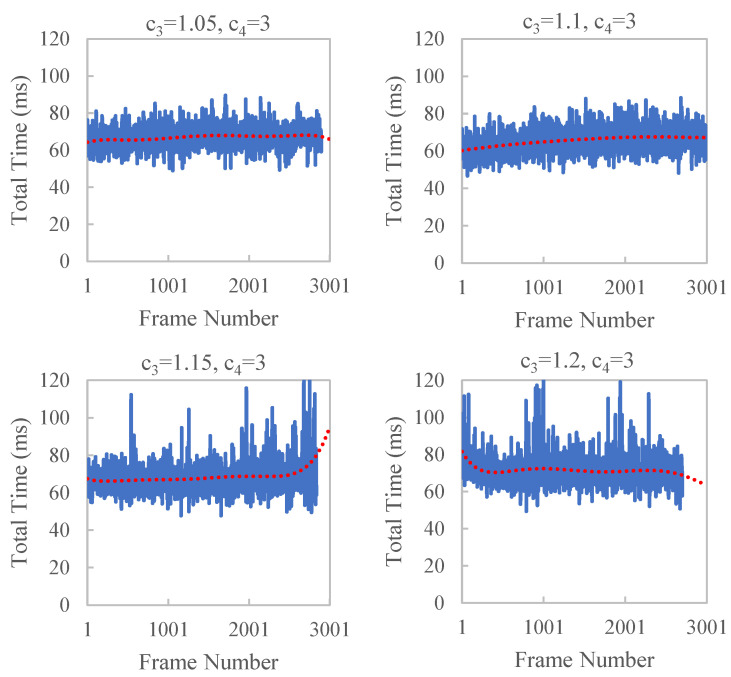
Time efficiency of reconstruction KITTI odometry sequence 00 with different $c_{3}$. When $c_{3}$ is larger than 1.1, large jitters in the running time occasionally appear.

**Figure 11 sensors-23-01876-f011:**
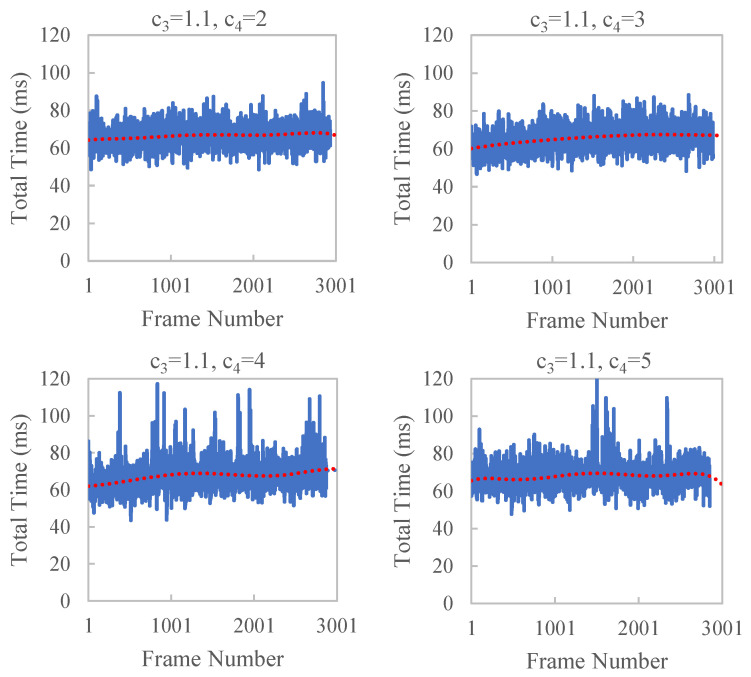
Time efficiency of reconstruction KITTI odometry sequence 00 with different $c_{4}$. When $c_{4}$ is larger than 3, large jitters in the running time occasionally appear.

**Table 1 sensors-23-01876-t001:** Reconstruction accuracy on ICL-NUIM (cm).

Methods	kt0	kt1	kt2	kt3
ElasticFusion	0.7	0.7	0.8	2.8
BundleFusion	0.5	0.6	0.7	0.8
FlashFusion	0.8	0.8	1.0	1.3
Dense Surfel Mapping	0.7	0.9	1.1	0.8
Ours	0.4	1.0	0.8	0.8

**Table 2 sensors-23-01876-t002:** Time efficiency (average).

SP_SIZE	FAR_DIST (m)	Generate Superpixels (ms)	Fusion (ms)	Total (ms)
8	10	35.6	1.3	38.8
8	20	56.1	1.4	59.9
8	30	60.5	1.4	64.7
4	10	37.1	1.4	41.2
4	20	60.8	2.0	67.6
4	30	63.4	2.1	70.6
8 [3]	30	≈70.0	≈1.0	≈75.0

**Table 3 sensors-23-01876-t003:** Time efficiency with different $c_{3}$ and $c_{4}$.

$c_{3}$	$c_{4}$	Average Time (ms)	Standard Deviation
1.1	2	66.5	5.3
1.1	3	65.5	5.8
1.1	4	67.4	7.1
1.1	5	68.0	6.4
1.05	3	66.9	5.2
1.15	3	68.2	7.3
1.2	3	71.5	7.3

## Data Availability

Not applicable.

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
