# Peer review of "A Dense Mapping Algorithm Based on Spatiotemporal Consistency"

_sensors, 2023, doi:10.3390/s23041876_

Round 1

Reviewer 1 Report

In this manuscript, a dense mapping algorithm is proposed based on Dense Surfel Mapping [3]. Although the base method [3] is introduced and difference between this paper and [3] is described in the introduction, the contribution of the paper compared with [3] is still vague. Moreover, description of the proposed method is ambiguous because what was already proposed in [3] is not clearly distinguished in sections 3 and 4. Additionally, experiment does not show clear advantage in comparison with [3]. Therefore, the contents and descriptions should be thoroughly reconsidered from all of the aspects described above.

In the fifth paragraph in the introduction section, Dense Surfel Mapping’s drawbacks are described. But the description is unclear, and hence, significance of the changes proposed in this paper was not understood. One possible solution for the problem is to first explain Dense Surfel Mapping in more detail with clear basic definitions recalled from [A] in section 3, and to explain [3]’s drawbacks, modifications proposed in this paper and significance of the modifications in more detail after that.

Description of the proposed method is very unclear because contents of Dense Surfel Mapping and proposed modification in this paper are mixed in section 3 and 4, while basic notations, definitions and methodologies are all omitted. Because of the mixture and disregard of fundamentals, the original contribution of the paper is incomprehensible.

Evaluation of performance of the proposed method in experiment is insufficient. Comparison of reconstruction accuracy in Table 1 does not make sense because most results are the same as shown in Table I in the paper [A]. Accuracy is almost the same as Dense Surfel Mapping except for the case of sequence kt0. Memory consumption shown in Figure 8 shows that the proposed method uses much larger size of memory than [3]. Since the proposal of the paper is some modifications from [3], superiority of performance in comparison with [3] is indispensable. Otherwise, all the modifications could be regarded as meaningless.

[A] Real-time Scalable Dense Surfel Mapping, arXiv:1909.04250v1 [cs.RO]

The followings are relatively minor comments.

Line 13, SLAM is not `Synchronous’ Localization and Mapping.

Equation (1), || p1 – p2 || should be used since p1 and p2 are vectors.

\cdot is used in Equation (2) to express inner product. But it is used for matrix multiplication in other equations, which should be avoided.

Figure 1 seems to be similar to Fig. 2 in [A], but the difference is not clear. It is recommended to use the same expression as [A] and add the authors’ proposal to the figure to show the proposed modification more clearly.

Line 110, `the other is the pose graph obtained by the sparse SLAM or other pose estimation algorithms (e.g. ORB-SLAM series[5,23,24], VINS-Mono[6], VINS-Fusion[25]). But what `the sparse SLAM’ means is not clarified or no reference is indicated.

This sentence seems to have been taken from “We use a state-of-the-art sparse visual SLAM

system (e.g. ORB-SLAM2 [18] or VINS-MONO [19])” from [3], but its meaning is different and incomprehensible.

Line 115, what detects loop closure? The description of system overview is incomplete in many aspects.

Line 117, definition of T1 and T2 are incomplete. Are they homogeneous transformation matrices?

Figure 2, generally, RGB-D sensor has restriction of view range not only in the sense that it cannot observe distant objects but also in the sense that it cannot detect objects that are too close. Isn’t is needed to consider closeness as well?

Description of 4.1, which seems to be original part of the method, is mathematically weak. 4.1.2 is very ambiguous only with explanation by sentence.

Equation (6), what is the size of the camera intrinsic matrix K?

Line 155, definition of range V2x1 should be given more clearly.

Equation (11)(12), is it correct/appropriate to apply the same transformation to the position (coordinate) and normal?

Equation (15), the intention of the equation is incomprehensible due to lack of abstract description.

Equation (17), \leftarrow should be used instead of `=’ if substitution is meant. Especially, the last two equations do not make sense because it simply means X = min(X,X).

Usage of ‘A ? B : C’ is not common and should be avoided.

Equation (19), isn’t is necessary to use || x ||?

Reviewer 2 Report

I have reviewed the paper entitled "A Dense Mapping Algorithm Based on Spatiotemporal Consistency" which proposes a local map extraction and fusion strategy based on spatiotemporal consistency.

I will suggest minor revision for this paper as it needs to be improved. My following suggestions and comments can help you to improve the paper.

1. In section 1, make an improved comparison between existing research work and your contribution

2: Section 2 of system overview is over simplified and lack the analytical details. Add some more technical depth to enhance the quality of the paper.

3: The results provided in this paper need to further explain and provide the further reasoning for the performance upgrade. 

4. Summarize the research contribution in section 5 with more quantitively manner. 

5. Do extensive revision for the English grammar and simplify the longer sentences to increase the readability of the paper.

6. Add reference from the 2022 research papers too.
